# Stability, Morphology, and Effects of In Vitro Digestion on the Antioxidant Properties of Polyphenol Inclusion Complexes with β-Cyclodextrin

**DOI:** 10.3390/molecules27123808

**Published:** 2022-06-14

**Authors:** Sunday Ntuli, Machel Leuschner, Megan J. Bester, June C. Serem

**Affiliations:** 1Department of Anatomy, Faculty of Health Sciences, University of Pretoria, Private Bag X323, Arcadia, Pretoria 0007, South Africa; u15116833@tuks.co.za (S.N.); megan.bester@up.ac.za (M.J.B.); 2Department of Pharmacology, Faculty of Health Sciences, University of Pretoria, Private Bag X323, Arcadia, Pretoria 0007, South Africa; machel.leuschner@gmail.com

**Keywords:** polyphenols, antioxidant, antiglycation, encapsulation, beta-cyclodextrin

## Abstract

Polyphenols are inversely associated with the incidence of chronic diseases, but therapeutic use is limited by poor stability and bioaccessibility. Encapsulation has been shown to overcome some of these limitations. A selection of polyphenols (catechin, gallic acid, and epigallocatechin gallate) and their combinations were encapsulated in beta-cyclodextrin (βCD). Encapsulation was characterized and the thermal and storage stability was evaluated using the 2,2-azinobis (3-ethylbenzothiazoline-6-sulfonic acid) (ABTS) assay. The samples were then subjected to in vitro digestion using a simple digestion (SD) model (gastric and duodenal phases) and a more complex digestion (CD) model (oral, gastric, and duodenal phases). Thereafter, the chemical (oxygen radical absorbance capacity assay) and cellular (dichlorofluorescein diacetate assay in Caco-2 cells) antioxidant and antiglycation (advanced glycation end-products assay) activities were determined. Inclusion complexes formed at a 1:1 molar ratio with a high encapsulation yield and efficiency. Encapsulation altered the morphology of the samples, increased the thermal stability of some and the storage stability of all samples. Encapsulation maintained the antioxidant activity of all samples and significantly improved the antiglycation and cellular antioxidant activities of some polyphenols following SD. In conclusion, the formed inclusion complexes of βCD with polyphenols had greater storage stability, without altering the beneficial cellular effects of the polyphenols.

## 1. Introduction

Polyphenols constitute the largest group of physiologically active phytochemicals that have numerous beneficial health effects, including antioxidant, antiglycation, anti-inflammatory, anticancer, and antimicrobial effects [1,2]. Increased intake of dietary polyphenols has been associated with a low incidence of non-communicable diseases (NCDs) [3]. Polyphenols, due to their wide range of health benefits, are being incorporated into foods such as dairy products and bread [4,5]. The development and use of nutraceuticals rich in polyphenols have great potential for NCD prevention and treatment. This may also limit or delay the need for pharmaceutical intervention often associated with side effects. For example, the use of metformin for the treatment of glucose intolerance in metabolic syndrome is often associated with nausea and diarrhea [6,7,8].

The application of polyphenols is still limited by several factors such as poor water solubility, lack of stability in solution, thermo- and photosensitivity, astringency, and the ability to bind protein [9,10]. For example, the polyphenols catechin (CAT) and gallic acid (GA), despite excellent thermal stability (up to 120 °C), both undergo degradation by photo-oxidation as a result of ultraviolet (UV) exposure [11]. The stability of epigallocatechin gallate (EGCG) is adversely affected by pH changes, temperature, and UV exposure both in solution and powder, resulting in degradation through epimerization and oxidation [12,13]. The applications of resveratrol, a potent antioxidant with anticancer and cardio- and neuro-protective properties, are limited by poor solubility, bioavailability, and adverse effects [14].

Encapsulation technology has been used to overcome some of the limitations on the use of polyphenols. A wide range of encapsulation techniques such as lyophilization, spray-drying, coacervation, and emulsification with polymer wall materials, such as βCD, maltodextrin, pectin, and sodium alginate, suitable for different active compounds are available [15,16]. Particularly, molecular inclusion complexation with cyclodextrins (especially βCD and its derivatives) has produced notable outcomes for polyphenol encapsulation [17]. Lu et al. [18] improved the water solubility and antioxidant activity of the poorly water-soluble resveratrol through encapsulation with βCD and its hydroxypropyl derivative. Encapsulating CAT in βCD doubled its water solubility, increased thermal and pH stability, improved astringency, and significantly reduced degradation due to atmospheric and photo-oxidation [10,19].

For successful therapeutic applications, the low bioaccessibility of polyphenols limited by several factors, including poor chemical stability in gastrointestinal fluids, high protein-binding affinity, interactions with other nutrients, metabolism, and rapid excretion, has to be improved [20]. Wojtunik-Kulesza et al. [21] summarized the effects of in vitro digestion on polyphenols, and the general trend was a significant loss of polyphenols following intestinal digestion. During digestion, pure polyphenols are subjected to extensive bio-transformation by colon microbes that subsequently alters their antioxidant activities [22]. Albeit some gut microbial metabolites have been shown to have similar activity to the parent compound [22,23]. Therefore, it is necessary to find strategies to increase the stability and better understand the effects of processes such as encapsulation on the structural and biochemical properties of polyphenols, including polyphenol mixtures, following digestion in an endeavor to develop improved polyphenol-rich products.

Xu et al. [24], reported that with βCD encapsulation, the stability of EGCG was increased and subsequently the inhibition of osteoclastogenesis by EGCG was increased. Roy et al. [25] reported that βCD encapsulation of GA, EGC, and EGCG did not prevent the formation of dityrosine cross linkages associated with oxidative stress but did prevent protein oligomerization. No study has been undertaken to investigate the effects of βCD encapsulation on the antioxidant properties of GA, CAT, and EGCG, especially as mixtures. The stability and bioactivity following digestion is an important consideration related to the health benefits of polyphenols. Grgić et al. [26] identified two studies that investigated the release, ex vivo permeation, and pharmacodynamics of CAT, in phosphatidylcholine liposomes, and the in vivo bioactivity retention and antidiabetic properties of CAT encapsulated in horse chestnut, water chestnut, and lotus stem starch with freeze drying. In addition, two studies investigated the retention/release of EGCG encapsulated in gum arabic maltodextrin and the stability of EGCG in chitosan-triphosphate in the mouse GIT.

For mixtures that contain the above polyphenols such as green tea or green tea polyphenols, different encapsulation methods, but not βCD, have been used and the effects of in vitro digestion on epithelial permeability and intestinal transport have been investigated. The effects of in vitro digestion on the antioxidant properties, including the cellular effects, are lacking, and consequently, in this study, the effect of in vitro digestion on the antioxidant properties of βCD-encapsulated GA, CAT, and EGCG alone and as part of mixtures was determined.

## 2. Results and Discussion

### 2.1. Confirmation of Inclusion Complexation Using Electrospray Ionization Tandem Mass Spectrometry (ESI-MS/MS)

The chemical identity of the inclusion complexes was elucidated using tandem mass spectrometry. The mass spectra (signal intensity against mass to charge ratio (*m*/*z*)) for both the encapsulated complex and the βCD and each polyphenol were assessed to confirm inclusion complexation. The average molecular weight of the inclusion complex for each polyphenol was calculated based on the average molecular masses of each compound. After qualitative identification of each βCD-polyphenol inclusion complex was obtained with Q_1_ scans, enhanced resolution scans were obtained and used to set the parent ion *m*/*z* to be scanned for in Q_1_. The tandem mass spectra for the product ions (fragmented in Q_2_ and isolated in Q_3_) of each parent ion were obtained and enhanced product ion scans were used to determine the product ion *m*/*z*. The molecular weight for the βCD-polyphenol inclusion complexes, individual polyphenols, and βCD with their Q_1_ and Q_3_
*m*/*z* and optimized mass spectrometry parameters are presented in Table 1.

The inclusion complexation of CAT with βCD was confirmed by the identification of the parent ion [CAT+βCD-H]^−^ (*m*/*z* 1424.5) and its product ions [βCD-H]^−^ (*m*/*z* 1134.4) and [CAT-H]^−^ (*m*/*z* 289.1) using ESI-MS/MS (Appendix A). The relative peak intensities of the parent [CAT+βCD-H]^−^ ion and the resulting product ions after relative collision-induced dissociation (CID) (applied in Q_2_) confirmed the 1:1 molar ratio for the CAT and βCD inclusion complex. Similarly, CAT inclusion complexation with βCD has previously been confirmed using ESI-MS. The inclusion complex ion was identified with a *m*/*z* of 1423 and 1133 *m*/*z* for free βCD in the full mass spectra [27]. Budryn et al. [28] also confirmed the inclusion complexation of βCD with different chlorogenic acids using the ESI-MS/MS method and observed a 1:1 molar ratio.

The inclusion complexation of GA with βCD at a 1:1 molar ratio was confirmed by the identification of the parent ion [GA+βCD-H]^−^ (*m*/*z* 1304.5) and its product ions [βCD-H]^−^ (*m*/*z* 1134.4) and [GA-H]^−^ (169.0 *m*/*z*) using ESI-MS/MS (Appendix A). da Rosa et al. [29] employed spectral, thermo-gravimetric, and differential scanning calorimetry tools to confirm the inclusion complexation of GA with βCD. During the formation of inclusion complexes, chromophores of the guest compound are shielded in the hollow cavity of cyclodextrin, resulting in alterations in the spectral profiles as seen in the Fourier transform infrared (FTIR) and ultraviolet-visible (UV-Vis) spectrometry [30]. The UV-Vis spectral changes observed for GA are consistent with the movement of the polyphenol into the βCD cavity [25,29]. Similarly, the transmittance changes seen in the FTIR spectra of GA following βCD encapsulation suggest strong interactions between the molecules [29].

The inclusion complexation of EGCG with βCD at a 1:1 molar ratio was confirmed by the identification of the parent ion [EGCG+βCD+H]^+^ (*m*/*z* 1594.2) and its product ions [βCD+H]^+^ (*m*/*z* 1135.2) and [EGCG+H]^+^ (459.0 *m*/*z*) using ESI-MS/MS (Appendix A). Interestingly, ions with *m*/*z* of 1424.3 and 289.1, similar to the parent ion of the CAT+βCD complex and CAT, respectively, were identified in the spectrum. This is most likely a result of the GA moiety breaking off from EGCG, thus leaving the ions of a CAT isomer and inclusion complexes of the isomer with βCD that have similar *m*/*z*, respectively. Roy et al. [25] confirmed the inclusion complexation of EGCG with βCD by studying the UV-Vis and proton nuclear magnetic resonance (^1^H NMR) spectrometry of the two compounds before and after encapsulation. The ^1^H NMR spectrometry ascertains inclusion complexation through the study of the spatial proximity of βCD protons with the guest compounds’ protons [31], and apparent changes in the chemical shifts for particular βCD protons following inclusion complexation are observed [30].

### 2.2. Encapsulation Yield and Efficiency

Using the inclusion complexation method with subsequent lyophilization, the recovered encapsulated CAT, GA, EGCG, and triple combination powders ranged from 91.27 ± 2.90–98.96 ± 8.89% (Table 2). The encapsulation yield was not significantly (*p* > 0.05) different between the individual encapsulated polyphenols and the combinations. Lyophilization has been shown to produce high encapsulation yields for polyphenol inclusion complexes with βCD [32,33]. Ho et al. [19] reported up to an 89% encapsulation yield for CAT encapsulated in βCD with lyophilization. A further 90.76 ± 0.58% encapsulation yield was reported for inclusion complexes of CAT with βCD obtained through co-precipitation with subsequent lyophilization [34]. These values are similar to the 91.27 ± 2.90% reported for the CAT and βCD inclusion complexes found in the current study. Other polyphenols, such as hydroxytyrosol, encapsulated in βCD through lyophilization also had encapsulation yields of up to 91.0% [33]. Moreover, considering the polyphenol-rich extract of blueberry (*Vaccinium myrtillus*) juice as an example for the combination samples, high encapsulation yield values of up to 78.1% have been previously reported [32].

Once the active compound is encapsulated, it is important to determine the efficiency of the encapsulation for the method used. In this study, the encapsulation through molecular inclusion complexation with subsequent lyophilization presented a high encapsulation efficiency (Table 2). Using this method, up to a 96.63 ± 0.40% encapsulation efficiency was determined for CAT. Ho et al. [19] reported an encapsulation efficiency of 94 ± 2.19% for CAT encapsulated in βCD, similar to the findings of the present study. However, a lower encapsulation efficiency has been documented, where Żyżelewicz et al. [27] reported only 61.5 ± 0.2 to 64.9 ± 0.3% and Jiang et al. [34] reported a 70.37 ± 1.02% encapsulation efficiency for inclusion complexes obtained through co-precipitation. For GA, an 80 ± 1.4% encapsulation efficiency has previously been reported [29]. Using a βCD derivative 2-hydroxypropyl-β-cyclodextrin (HPβCD) with subsequent spray drying, an encapsulation efficiency of 89.22% for GA encapsulated in HPβCD was reported [35]. The encapsulation efficiencies of GA observed in previous studies are lower than the 95.57 ± 1.57% reported in the current study. While several studies have investigated the encapsulation of EGCG with βCD, no reported encapsulation efficiency could be found [25,31,36].

The encapsulation efficiency was significantly higher for co-encapsulated polyphenol samples. Similarly, the co-encapsulation of curcumin with resveratrol [37] and vitamin C with β-carotene [38] in liposomes resulted in an increased encapsulation efficiency. However, Olga et al. [35] reported a significant decrease in the encapsulation efficiency for GA co-encapsulated with ferulic acid in HPβCD while it remained unchanged for encapsulated ferulic acid. The encapsulation efficiency is affected by factors such as the concentration and solubility of the guest and host compounds, the binding constants, and the size of the guest compound relative to the host cavity [35]. Hence, the adsorption (translated to encapsulation efficiency) onto gelatin nanoparticles was higher for polyphenols with higher molecular weights and more hydroxyl groups [39]. This is further evident in the association constants reported for GA, epicatechin gallate (ECG), and EGCG, where the values increased with the number of hydroxyl groups on respective compounds [25]. In the current study, the encapsulation efficiency increased with the molecular weight and number of hydroxyl groups in each polyphenolic sample, from the smallest polyphenol GA to the CAT/GA/EGCG triple combination.

### 2.3. Surface Morphology

The surface morphology of the polyphenols following inclusion complexation was studied using scanning electron microscopy (Figure 1). All tested compounds presented a crystalline structure. The βCD crystals (Figure 1a) had irregular shapes and an orientation (blue arrow) consistent with previous reports [19]. The crystals of CAT (Figure 1b) were needle-shaped (light green arrow), similar to previous studies [19,34]. The gallic acid micrographs (Figure 1c) presented uniform crystalline rods (purple arrow) similar to that identified by da Rosa et al. [29]. The crystals of EGCG (Figure 1d) were long, flat, and thin (red arrow) as previously described by Cao et al. [40]. The mixtures presented morphologies characteristic of each polyphenol in the sample (Figure 1e).

Following lyophilization, βCD (Figure 1f) presented an amorphous morphology (green arrow). Similarly, polyphenols lyophilized without encapsulation also presented an amorphous morphology (data not shown). The inclusion complexes obtained through lyophilization (Figure 1g–j) presented a crystalline lamellate morphology (orange arrow) consistent with previous reports [5,34]. The original morphology of the pure compounds was not observed in all the inclusion complexes; this observation was considered as confirmation of inclusion complex formation as identified in previous studies [19,29,34].

### 2.4. Thermal Stability

The poor thermal stability of polyphenols is well documented [13,19]. The polyphenol samples tested in the present study retained antioxidant activity following thermal treatment (Figure 2).

The % antioxidant retention was significantly (*p* < 0.05) higher for inclusion complexes of βCD with CAT relative to free CAT following 1 and 5 h of incubation at 100 °C. After 5 h, a 39.11 ± 4.36 and 25.27 ± 5.46% loss of antioxidant activity was observed for free and encapsulated CAT, respectively (Figure 2a). Jiang et al. [34] reported an antioxidant retention of up to 69.88 ± 3.90 and 87.24 ± 2.11% for free and βCD complexed CAT, respectively, following 1 h of incubation at 100 °C. Similarly, in the present study, after 1 h of incubation at 100 °C, the antioxidant retention of free and encapsulated CAT was 71.41 ± 3.76 and 81.16 ± 2.97%, respectively. In both studies, the antioxidant retention of encapsulated CAT was significantly higher than that of non-encapsulated CAT.

For GA, the % antioxidant retention was significantly (*p* < 0.05) higher at 3 h for free GA compared with the encapsulated GA. After 5 h, the free and encapsulated GA lost 31.65 ± 10.04 and 28.37 ± 7.58% antioxidant activity, respectively (Figure 2b). Inclusion complexes of GA did not exhibit significant antioxidant retention compared with free GA. Similarly, Volf et al. [11] reported up to 30% thermal degradation of GA quantified using HPLC after 4 h of incubation at 100 °C. The lack of thermal protection by encapsulation is comparable with the thermograms obtained by da Rosa et al. [29] for GA inclusion complexes with βCD. At temperatures less than 325 °C, there was a lack of differences between the curves of the physical mixtures and the inclusion complexes of GA with βCD, thus indicating a lack of thermal protection [29].

For EGCG at 4 and 5 h, the % antioxidant retention was significantly (*p* < 0.05) higher for the inclusion complexes compared with free EGCG. In addition, after 5 h of incubation, 36.10 ± 7.51 and 16.96 ± 4.02% activity was lost for free and encapsulated EGCG, respectively (Figure 2c). The cis-configured catechins are thermosensitive and up to 75% of EGCG has been reported to be lost to thermal degradation following heat treatment at 120 °C [12]. In the present study, the amount of antioxidant activity lost at 4 and 5 h for free EGCG was at least 2-fold higher compared with EGCG inclusion complexes with βCD, indicating apparent thermal protection. In a similar study, derivative thermogravimetric analysis of EGCG encapsulated in chitosan or gelatin also revealed an improvement in thermal stability [41].

For the double combinations, no significant differences were observed between the free and encapsulated samples (data not shown). The triple combination inclusion complexes had significantly higher activity compared with the free combination sample after 1 h of incubation. After 5 h of incubation, the free and encapsulated CAT/GA/EGCG triple combination lost up to 36.39 ± 3.81 and 32.88 ± 6.54% activity, respectively (Figure 2d). The observed loss of antioxidant activity of the combination samples due to thermal treatment is consistent with previous findings. The polyphenol-rich extracts of açaí and maqui berry fruits lost about 10% phenolic content following thermal treatment at 121 °C [42,43]. In the present study, inclusion complexation generally did not provide thermal protection to the polyphenol mixtures. However, HPβCD encapsulation provided thermal stability for maqui berry polyphenol extract, where over 90% of the phenolic content quantified with the Folin–Ciocalteu (FC) assay was retained. This observation was further supported by thermogravimetric studies [43].

### 2.5. Storage Stability

The polyphenolic samples retained antioxidant activity following exposure to atmospheric oxygen for two weeks (Table 3). For the control, non-encapsulated, and encapsulated samples, the antioxidant activity was similar. After one week, comparing the non-encapsulated with the encapsulated samples, the antioxidant activity was significantly reduced for CAT/GA/EGCG. After 2 weeks, the antioxidant activity was decreased for the non-encapsulated samples compared with the encapsulated samples and this was significant for CAT, GA, and EGCG. For all other samples, after 1 and 2 weeks, the antioxidant activity of non-encapsulated and encapsulated samples was unchanged.

After two weeks of exposure to atmospheric oxygen, compared with the control, antioxidant activity was significantly reduced for the non-encapsulated and encapsulated CAT, GA, EGCG, and CAT/GA/EGCG, and was reduced for most of the double combinations. The greatest reduction in antioxidant activity was observed for non-encapsulated samples compared to the control at week 2.

Several studies have reported the oxidation of CAT following exposure to air [19,34]. In this current study, up to 68.75 ± 1.94% CAT antioxidant activity was retained following exposure to air for 14 days. This is slightly higher compared with previous studies, where after 14 days, the reported antioxidant retention for CAT was about 30% [19,34]. In the present study, amorphous powders were used, and these provide a higher retention. Nevertheless, there is a consistent loss of antioxidant activity by CAT when exposed to air [19,34]

Phenolic acids such as caffeic-, p-coumaric-, and ferulic acid in coffee have been shown to be sensitive to atmospheric oxidation. The antioxidant activity of regular and decaffeinated coffee rich in these phenolic acids was 50.1 and 56.6% after 6 months of exposure to air at 20 °C [44]. In the current study, the antioxidant retention by GA significantly dropped to 87.61 ± 0.86% following 2 weeks of exposure to air at room temperature and was significantly improved (96.31 ± 0.48%) by inclusion complexation with βCD. In another study, the stability of GA (quantified using the FC assay) encapsulated in a complex of whey protein concentrate and pectin polysaccharide was more than 90% after 2 weeks of storage at 20 °C [45].

In both powder and solution formulations, EGCG is similarly affected by low oxidative stability [46,47]. In the current study, the EGCG antioxidant retention was 77.50 ± 0.62% after 2 weeks of exposure to air, and with encapsulation this increased to 85.44 ± 3.83%. Zhu et al. [47] reported antioxidant retention of at least 80% for non-encapsulated EGCG after 2 weeks of storage at 30 °C with exposure to air. Encapsulation of EGCG in cassava starch then improved the antioxidant retention of EGCG to at least 91% [47].

The combination samples were generally impervious to atmospheric oxidation (data not shown). Of note, there was a significant increase in the antioxidant retention for encapsulated CAT/GA (145.26 ± 3.04%—week 1, 140.88 ± 4.63%—week 2). This is comparable with the increase seen in the antioxidant activity of rutin and mesquitol solutions, where Chaaban et al. [48] postulated that the degradation products had higher antioxidant activity relative to native forms and the increase in activity could be attributed to synergistic effects between neoformed and native molecules. Although the antioxidant retention of the encapsulated combination samples was generally higher than non-encapsulated samples following exposure to air, the differences were not statistically significant. A similar trend was noted in thermal stability studies, where inclusion complexation did not offer significant protection to the combination samples against treatment.

### 2.6. Antioxidant Activity–Oxygen Radical Absorbance Capacity (ORAC)

The ORAC assay measures the free radical quenching antioxidant capacity of compounds through proton transfer [49]. The tested samples showed antioxidant activity through hydrogen atom transfer, a known mechanism for polyphenols, as assessed using the ORAC assay (Figure 3).

For free CAT, GA, and EGCG samples, the antioxidant activity pre-digestion (ND) was 130.50 ± 6.36, 104.84 ± 4.93, and 189.23 ± 12.53 µM TE, respectively (Figure 3). The CAT activity is similar to the 129.2 ± 1.2 and 149.0 ± 8.0 µM TE previously reported by Żyżelewicz et al. [27] and Dávalos et al. [50], respectively. In addition, the activity quantified for GA is similar to the 111 ± 5.8 and 130.06 ± 24.46 µM TE reported by Zulueta et al. [51] and Roy et al. [52], respectively. For EGCG, the activity is similar to the 182.67 ± 10.26 µM TE reported by Zhou et al. [53].

Following encapsulation, the antioxidant activity of the non-digested polyphenols was maintained with 135.25 ± 5.06, 105.87 ± 6.73, and 199.95 ± 9.16 µM TE for encapsulated CAT, GA, and EGCG, respectively (Figure 3). In contrast, studies have reported an increase [19] or decrease [27,54] in the activity for CAT encapsulated in βCD and its derivatives. For GA, Hu et al. [55] reported a significant (*p* < 0.05) decline in AAPH scavenging activity following conjugation with chitosan. For EGCG, Folch-Cano et al. [54] reported an increase in the ORAC index of EGCG following inclusion complexation, similar to the findings of this study, where EGCG showed a slight but not statistically significant (*p* > 0.05) increase in activity.

Conditions associated with GIT digestion caused GA and EGCG to slightly lose antioxidant activity and similar findings have been reported for GA [56] and EGCG [57]. The study by Oliveira and Pintado [58] reported preservation and/or improvement of the AAPH scavenging activity of CAT encapsulated in β-lactoglobulin and pectin or chitosan following in vitro gastrointestinal digestion. In the current study, βCD encapsulation preserved the AAPH scavenging activity of CAT and GA following in vitro digestion, with non-encapsulated GA showing a significant decrease in activity (from 104.84 ± 4.93 to 69.19 ± 5.73 µM TE) following SD. With CD, both free and encapsulated EGCG showed a decrease in antioxidant activity from 189.23 ± 12.53 to 151.53 ± 13.60 µM TE and 199.95 ± 9.16 to 159.82 ± 7.12 µM TE, respectively (Figure 3). Similarly, the AAPH radical scavenging activity of both free and broccoli by-product puree and pomace complexed EGCG was significantly (*p* < 0.05) reduced following a modified complex in vitro digestion [57]. In contrast, the complexes of EGCG with whey protein isolate had increased AAPH scavenging activity following in vitro simulated digestion, where free EGCG lost activity [59].

For the non-digested, non-encapsulated CAT/GA/EGCG combination (Figure 3), the antioxidant activity was 233.49 ± 11.36 µM TE. The antioxidant activity was maintained following encapsulation at 240.30 ± 10.11 µM TE. Similar observations were made with the double combination samples.

All samples had antagonistic interactions that were maintained following encapsulation *p* < 0.001, except for CAT/GA, which had additive interactions pre and post encapsulation (Appendix A). Freeman et al. [60] studied the individual interactions of polyphenols from navel oranges, including chlorogenic acid, hesperidin, luteolin, myricetin, naringenin, p-coumaric acid, and quercetin, at concentrations found in the oranges (0.786–10.7 µM) and different double and triple combinations exhibited synergistic, additive, and antagonistic interactions towards AAPH radical scavenging. Furthermore, Carbonneau et al. [61] reported antagonistic interactions between the two common sorghum 3-deoxyanthocyanidins, luteolinidin and apigeninidin. These findings are consistent with the results of the present study, where both additive and antagonistic interactions were observed depending on the polyphenols present in each combination sample.

Complex in vitro digestion significantly (*p* < 0.05) decreased the ORAC values of the non-encapsulated and encapsulated combination sample, and together with the results observed for the encapsulated simple digested samples, the conclusion is that in the present study, encapsulation did not provide protection against the GIT digestion-mediated degradation. Polyphenol instability at neutral or slightly alkaline pH may account for the loss of antioxidant activity as described for tea polyphenols [12,62]. The longer exposure times for CD may account for the greater loss of activity compared with SD.

### 2.7. Antioxidant Activity–Advanced Glycation End-Products (AGEs)

The AGEs assay was used to quantify the ability of polyphenols to inhibit the glycation of proteins in vitro. The inhibition of bovine serum albumin (BSA) glycation in the BSA-methylglyoxal (MGO) and BSA-fructose (FRU) models by the polyphenol samples is presented in Figure 4.

The antiglycation activity of free CAT, GA, and EGCG in the BSA-MGO model before digestion (ND) was 18.95 ± 1.38, −2.31 ± 1.31, and 25.90 ± 1.32%, respectively (Figure 4a). The antiglycation activity of CAT was comparable with previous reports [63,64,65], where at least 20% AGEs formation was inhibited at concentrations similar to that of the present study. The lack of antiglycation activity in the BSA-MGO model observed for non-encapsulated and encapsulated GA is consistent with the findings of de Lima-Júnior et al. [64]. In contrast, Park and Lee [65] and Spagnuolo et al. [66] demonstrated that at relatively high concentrations, GA exhibited antiglycation activity in a dose-dependent manner. Both EGCG and its isomer gallocatechin gallate (GCG) have strong dose-dependent antiglycation activity [67,68]. Furthermore, EGCG at 100 µM had up to 69.1% antiglycation activity [69], more than twice the 25.90 ± 1.32% reported in the present study at the same concentration. This difference could, however, be attributed to some minor differences in the BSA-MGO model method employed in respective studies.

Encapsulation of polyphenolic delphinidin and cyanidin in nanoliposomes have previously been observed to significantly improve in vitro antiglycation activity in a BSA-glucose model. In vivo studies further revealed improved antiglycation activity as a result of encapsulation [70]. However, the encapsulation of grape skin polyphenols in alginate microbeads resulted in a decrease in the antiglycation activity in an in vitro BSA-MGO model [71]. In the present study, inclusion complexation retained the antiglycation activity of all polyphenols, with no significant differences observed between non-encapsulated and encapsulated samples. The antiglycation activity of non-digested encapsulated CAT, GA, and EGCG was 12.51 ± 3.60, −4.23 ± 3.73, and 21.69 ± 3.53%, respectively.

The antiglycation activity of the non-encapsulated triple combination in the BSA-MGO model before digestion was 38.02 ± 1.30%. With encapsulation, the antiglycation activity was 33.63 ± 2.75% prior to in vitro digestion for CAT/GA/EGCG. Data for the double combination samples is presented in Appendix A. Considering extracts as examples for polyphenol mixture samples, several polyphenol-rich plant extracts have been previously shown to exhibit antiglycation activity [72,73,74,75]. The antiglycation activity of the non-encapsulated combinations exhibited antagonistic interactions and changed to additive interactions with regards to the antiglycation activity following encapsulation (Appendix A). In the present study, inclusion complexation improved polyphenol–polyphenol interactions of the CAT/EGCG and CAT/GA/EGCG combinations from antagonistic to additive interactions. Conversely, the polyphenol–polyphenol interactions of GA/EGCG changed from synergistic in the non-encapsulated samples to additive following inclusion complexation.

For non-encapsulated CAT and GA, a significant increase in the antiglycation activity was observed following SD and CD while the activity of EGCG was unchanged. With encapsulation, a significant increase in activity was observed for CAT and GA following complex digestion, and encapsulated EGCG following simple digestion. For CAT/GA/EGCG, the activity was only increased after complex digestion, with no differences between non-encapsulated and encapsulated samples (Figure 4a). Only for EGCG after simple digestion did encapsulation increase the antiglycation activity. Complex in vitro digestion has been observed to significantly reduce the antiglycation activity of polyphenol-rich extracts of *Elaeagnus umbellata* and *Sambucus lanceolate* [76] and further *Rumex maderensis* [75], in a BSA-ribose in vitro model. In this study, an improvement in the antiglycation activity for both non-encapsulated and encapsulated samples was observed following in vitro digestion, especially after complex-simulated digestion. Following simple digestion, non-encapsulated CAT had significantly higher activity compared with encapsulated CAT.

In the BSA-FRU model (Figure 4b), the antiglycation activity of non-encapsulated CAT, GA, and EGCG was 47.31 ± 3.95, −5.67 ± 7.29, and 93.85 ± 2.54%, respectively. Similar antiglycation activity of 51.22 ± 2.90, −2.88 ± 3.08, and 92.66 ± 1.68% for encapsulated CAT, GA, and EGCG, respectively, was observed. The antiglycation activity of both non-encapsulated and encapsulated CAT was significantly higher in the BSA-FRU model compared with the BSA-MGO model. Significantly higher CAT antiglycation activity in the BSA-FRU model compared with the BSA-MGO model has been previously reported [63,64,65]. Furthermore, the antiglycation activity of CAT was significantly higher in a lysine and MGO (LYS-MGO) model and lower in an arginine and MGO (ARG-MGO) model compared with the BSA-MGO model at the same sample concentrations [64]. The findings in the present study, where free and encapsulated GA did not exhibit antiglycation activity in the BSA-FRU model, are consistent with the findings of Park and Lee [65]. Conversely, de Lima-Júnior et al. [64] observed significant antiglycation activity for GA in the BSA-FRU model compared with the BSA-MGO model and further inhibition was reported in the LYS-MGO and ARG-MGO models. Wu et al. [77] previously reported antiglycation activity for EGCG in the BSA-MGO model that was comparable with the results of the present study. Furthermore, the antiglycation activity of EGCG was significantly higher in the BSA-FRU model compared with the BSA-MGO model. Other flavonoids such as quercetin have similarly been reported to exhibit significantly higher antiglycation activity in the BSA-FRU model when compared with the BSA-MGO, LYS-MGO, and ARG-MGO model by de Lima-Júnior et al. [64].

For the non-encapsulated CAT/GA/EGCG combination, the antiglycation activity in the BSA-FRU model was 80.84 ± 4.73%, and with encapsulation, it was mostly unchanged at 75.42 ± 1.57%; this was higher than in the BSA-MGO model. The data for the double combination samples is presented in Appendix A. Similarly, polyphenol-rich extracts of *Syzygium cumini* [74], peanut skin and grapes [73], and brown algae [65] were shown to exhibit higher antiglycation activity in the BSA-FRU than the BSA-MGO model. The combination samples all exhibited antagonistic interactions towards the inhibition of BSA glycation by FRU (Appendix A). This observation could largely be attributed to the lack of antiglycation activity by GA and the almost total inhibition of FRU-induced BSA glycation by EGCG. Combinations of EGCG and epicatechin gallate at 1:2 and 2:1 mole ratios had mostly antagonistic interactions towards the inhibition of FRU-induced BSA glycation. However, at 1:1 mole ratios, more additive and some synergistic interactions were observed [77]. Chen et al. [78] also previously demonstrated that sample ratios in combinations play a significant role in obtaining the maximum antiglycation activity of polyphenols.

Following simple in vitro simulated digestion, the antiglycation activity was higher in the BSA-FRU than in the BSA-MGO model. However, simple digestion caused a significant reduction in the antiglycation activity of the encapsulated combinations containing EGCG compared to non-digested controls (Appendix A). Further, where simple digestion showed improved antiglycation activity, it was still significantly lower in the encapsulated samples compared to the non-encapsulated polyphenols. This could be due to the absence of the digestive salts and other enzymes in the simple digestion that are used in the complex method and are present in vivo. Nevertheless, in vitro digestion has been observed to decrease the antiglycation activity of polyphenol-rich extracts using the BSA-FRU model [75,76]. Complex digestion resulted in total inhibition of FRU-induced BSA glycation for all but non-encapsulated and encapsulated GA samples. In the BSA-MGO model, complex digestion resulted in a significant increase in the antiglycation activity of the samples but to a lesser degree than that observed in the BSA-FRU model.

The higher antiglycation activity observed in the BSA-FRU than in the BSA-MGO model could be dependent on the inhibition pathway or the inhibitor type and the antiglycation model [65,79]. It is also important to consider that MGO is the most potent glycation agent and dicarbonyl compounds are more reactive than monosaccharides [80,81]. Furthermore, the concentration of the inhibitor plays a significant role in antiglycation activity regardless of the model used [64,65]. Nevertheless, the higher inhibition in the FRU model is more relevant when considering that the dietary fructose present in many food products can result in metabolic syndrome manifestations, including upregulation of AGEs formation [82].

### 2.8. Antioxidant Activity–Dichlorofluorescein Diacetate (DCFH-DA)

The DCFH-DA assay was used to assess the cellular antioxidant activity (CAA) of the non-encapsulated and encapsulated polyphenol samples before and after in vitro simulated digestion in Caco-2 cells (Figure 5).

At 10 µM polyphenols, the CAA was 30.68 ± 11.13% for CAT and 27.94 ± 8.25% for EGCG while no significant CAA was observed for GA. At 100 µM, the CAA for non-encapsulated CAT, GA, and EGCG was increased to 80.30 ± 3.28, 89.79 ± 1.46, and 100.40 ± 1.02, respectively. Similarly, Kellett et al. [83] reported a dose-dependent 45 and 55% CAA for CAT at the 10 and 100 µM concentrations using a slightly modified method. While GA did not exhibit any CAA at 10 µM in the present study, Wang et al. [84] reported significant CAA (just under 25%) for GA at the same concentration against tert-butylhydroperoxide (t-BOOH)-induced oxidation. Furthermore, the 89.79 ± 1.46% CAA observed for GA at 100 µM confirms the dose-dependent CAA of GA [84]. Considering EGCG, Wan et al. [85] reported up to 50% CAA for EGCG at 38.5 µM using a slightly modified method. Together with the CAA reported by Wan et al. [85], the results of the present study suggest a dose-dependent CAA for EGCG.

Inclusion complexation maintained the CAA of 10 µM of each polyphenol prior to simulated digestion with 21.15 ± 11.78 and 23.30 ± 12.13% for encapsulated CAT and EGCG, respectively, where GA inclusion complexes did not exhibit any CAA. At 100 µM, the CAA was similarly maintained following encapsulation with CAT, GA, and EGCG, exhibiting 90.95 ± 2.21, 94.24 ± 1.24, and 100.72 ± 0.52% for the non-digested samples, respectively. While the literature on the CAA of encapsulated CAT is limited, encapsulation of other polyphenols such as resveratrol in nanoemulsions has been shown to maintain the CAA [86]. Recently, Thiengkaew et al. [87] reported a significant increase in CAA for mangiferin following encapsulation in nanobilosomes. Considering GA, Yi et al. [88] recently reported 83.5% CAA for GA conjugated with chitosan, consistent with the 94.24 ± 1.24% reported for GA inclusion complexes in the present study. Regarding EGCG, inclusion complexation preserved its CAA at both 10 and 100 µM. Similarly, encapsulation of EGCG in starch retained its CAA while encapsulation in lecithin and β-glucan wall materials improved the activity [89].

The CAA of the 10 µM samples was lost following in vitro digestion, with no significant differences observed between non-encapsulated and encapsulated samples. At 100 µM, the samples were not significantly affected by in vitro digestion except for GA. Similarly, Sessa et al. [86] reported a lack of a significant effect on the CAA of resveratrol encapsulated in various nanoemulsions after simulated digestion. Non-encapsulated GA at 100 µM completely lost CAA following simple digestion while encapsulated GA was not affected, and both completely lost CAA after complex digestion. Similarly, the CAA of EGCG encapsulated in niosomes was significantly higher compared with non-encapsulated EGCG following in vitro digestion [90]. These results reveal the protective effects of encapsulation on CAA against the harsh conditions of the GIT.

The CAA for the 10 µM CAT/GA/EGCG combination was 75.60 ± 3.69% prior to simulated digestion, whereas at 100 µM, slightly higher CAA at 99.10 ± 0.49% was observed. The encapsulated 10 µM combinations had improved CAA from additive (non-encapsulated) to synergistic interactions for most samples, whereas for the 100 µM samples, majority additive interactions were observed (Appendix A). Phan et al. [91] and Phan et al. [92] studied the 1:1 mole ratio combinations of the carotenoids lutein and β-carotene, respectively, with different anthocyanin glucosides and reported additive interactions towards CAA in Caco-2 cells. In another study, combinations of lycopene with anthocyanin glucosides had antagonistic interactions at a 1:1 mole ratio and additive interactions at a 3:1 mole ratio [93].

In the study by Elisia and Kitts [94], up to 50% CAA was reported for 55.1 ± 2.4 and 6.5 ± 0.3 µg/mL of crude blackberry extract and the purified anthocyanin extract, respectively, in Caco-2 cells. These findings suggest additive and/or synergistic interactions by the purified anthocyanin extract; hence, a lower polyphenol concentration was required when compared with the crude extract. Further, in t-BOOH-induced oxidation using Caco-2 cells, polyphenol-rich extracts of apple peels [95] and *Viburnum opulus* [96] exhibited up to 30 and 20% CAA, respectively. These results indicate that synergistic interactions play an important role in the CAA of polyphenols.

Following encapsulation, the CAA was 80.20 ± 3.30% for 10 µM βCD inclusion complexes with CAT/GA/EGCG and at 100 µM, this increased to maximum protection at 100.58 ± 0.57%. Encapsulating grape marc extract polyphenols in a carbohydrate wall material maltodextrin maintained the CAA of the active compounds [97]. Similarly, encapsulating polyphenols in the carbohydrate βCD maintained the CAA and polyphenol–polyphenol interactions observed with the non-encapsulated samples. In contrast, encapsulation of grape marc polyphenols in various nanoemulsions [86] and anthocyanins in nanoliposomes [98] significantly enhanced the CAA in Caco-2 cells.

The combination samples were generally impervious to in vitro digestion. Only the free 10 µM GA/EGCG (data not shown) showed a significant decline in CAA following simple digestion and all 10 µM samples completely lost CAA after complex digestion. The 100 µM combinations remained unchanged after simple or complex digestion. Considering pine bark polyphenol extracts as an example of a polyphenol mixture sample, Ferreira-Santos et al. [99] reported a significantly higher CAA for extracts encapsulated in maltodextrin compared to their non-encapsulated counterparts following in vitro digestion.

### 2.9. Cytotoxicity

To ensure that the CAA concentrations of the polyphenol samples had no adverse effects on cell viability, the crystal violet (CV) assay was used to assess the potential cytotoxic effects. None of the polyphenol samples displayed cytotoxic effects on the Caco-2 cells (data not shown). Similarly, previous studies reported that non-encapsulated and encapsulated polyphenolic samples at concentrations that exhibited CAA in Caco-2 cells did not demonstrate any cytotoxic effects [83,89,91,97].

## 3. Materials and Methods

Beta-cyclodextrin (βCD), gallic acid (GA), catechin (CAT), and epigallocatechin gallate (EGCG) were purchased from Merck SA (Pty), Sandton, South Africa (SA). For digestion, salivary amylase, pepsin from porcine gastric mucosa, pancreatin from porcine pancreas, porcine bile extract, Pefabloc^®^ and associated salts, and the reagents for the antioxidant studies were purchased from the same company. For electrospray ionization tandem mass spectrometry and electron microscopy, reagents were analytical grade and were obtained from Merck SA (Pty), SA. Cell culture media was also obtained from Merck SA (Pty), SA and fetal calf serum (FCS) were obtained from Capricorn Scientific, GmbH, Germany, and the Caco-2 cell line was obtained from CELLONEX, Separations, SA.

### 3.1. Preparation of Inclusion Complexes

To encapsulate the polyphenols using the molecular inclusion complexation technique, a 1:1 molar ratio of guest (polyphenol) to host (βCD) was used with the method modified according to Ozdemir et al. [100]. Equimolar masses for the 8 mM βCD material and each polyphenol powder alone (GA, CAT, and EGCG) and in combination (GA/CAT, CAT/EGCG, GA/EGCGC, and CAT/GA/EGCGC) were weighed and added to a conical flask covered with aluminum foil to protect the contents from direct sunlight exposure. The βCD powder was added to obtain a final concentration of 8, 16, and 24 mM in the single, double, and triple combination samples, respectively, to maintain the 1:1 molar host to guest ratio. A volume of 10 mL of ddH_2_O was then added to the powders, sealed, and magnetically stirred at 200 rpm for 24 h at ambient temperature. The solutions were then filtered through a 0.2 µm filter and frozen at −80 °C for 24 h. The frozen samples were lyophilized for 48 h using the Labconco Freezone 6 freeze drying system (Labconco, Kansas City, MO, USA) to obtain the dry encapsulated powders. The powders were collected, weighed, and kept at −20 °C in the dark until analysis.

### 3.2. Confirmation of Inclusion Complexation Using Electrospray Ionization Tandem Mass Spectrometry

Confirmation of inclusion complexation was performed using an ESI-MS/MS method following Żyżelewicz et al. [27] with slight modifications. Encapsulation of the polyphenol samples with βCD was confirmed using a triple quadrupole tandem mass spectrometer equipped with a Turbo-V^®^ electrospray ionization source and data acquisition managed using the Analyst™ Software, version 1.5.2 (Applied Biosystems, Toronto, ON, Canada). For this purpose, 2 mg of each of the encapsulated powders was dissolved in 10 mL ddH_2_O. An aliquot of 1 mL of the resulting solution was diluted in 9 mL of either methanol:acetonitrile (MeOH:ACN) (80:20) with 0.1% formic acid or MeOH:ACN (50:50) with 10 mM ammonium formate for positive and/or negative ionization, respectively. Encapsulated samples were directly injected into the mass spectrometer at a 20 µL/min flow rate using a Harvard syringe pump with a 1 mL syringe. Mass spectrometric detection parameters, i.e., the declustering potential, collision energy, and collision cell exit potential, were optimized for each sample to identify the inclusion complex parent ion and its fragmented product ions (mass to charge ratio of each polyphenol and βCD) with high selectivity and sensitivity.

### 3.3. Encapsulation Yield and Efficiency

To determine the encapsulation yield, the encapsulated powders recovered after freeze drying were weighed to 4 decimal places and the encapsulation yield calculated using Equation (1):% EY = (m_1/_m_0_) × 100(1)
where EY is the encapsulation yield, m_1_ is the mass of the recovered encapsulated powders, and m_0_ is the mass of the starting raw materials.

The encapsulation efficiency was determined according to Ho et al. [19] and Liu et al. [101] with modifications. For this purpose, 2 mg of each inclusion complex was accurately weighed and washed with 1 mL of >99.9% ACN by vortex mixing for 2 min. The suspension was then filtered through a 0.2 µm syringe filter and the filtrate was sampled. In total, 2 mg of each of the encapsulated powders were re-suspended in 1 mL ddH_2_O to serve as a control for the total polyphenol content measured. The polyphenol content in the filtrate and/or re-suspended powders was determined using the FC assay.

The FC assay was used to determine the polyphenol content of the samples and filtrate with modifications according to Serem and Bester [102]. A 49.5 µL volume of FC reagent solution (diluted 15× in ddH_2_O) was added to 11 µL of the filtrate and/or the re-suspended encapsulated sample solutions in a 96-well plate. A further 49.5 µL of 7.5% (w/v) sodium carbonate solution was added to the mixture. The plate was then mixed well, and the absorbance was measured at 630 nm using the EMax^®^ Microplate Reader (Biochrom Ltd., Cambridge, England). A concentration range (0.0–1.0 mM) of CAT, GA, and EGCG (alone and in combinations) prepared in ddH_2_O was used to prepare standard curves for determination of the respective polyphenols in the filtrate and/or re-suspended solution. The encapsulation efficiency was calculated using Equation (2):% EE = ((measured TPC − measured free PC)/(measured TPC)) × 100(2)
where EE is the encapsulation efficiency, TPC is the total polyphenol content, and PC is the polyphenol content.

### 3.4. Morphological Characterization of the Inclusion Complexes

For the morphological analysis of the inclusion complexes, a sample of each powdered inclusion complex was sprinkled onto a two-sided conductive carbon adhesive tape. The powders were then coated with a thin layer of carbon under vacuum and visualized using the Zeiss Crossbeam 540 FIB-SEM (Carl Zeiss Microscopy, Jena, Germany) scanning electron microscope.

### 3.5. Thermal and Storage Stability of the Inclusion Complexes with Respect to the Antioxidant Retention

The thermal stability of the sample solutions was determined according to Mangolim et al. [103] and Ho et al. [19] with slight modifications. Sample solutions of 75 µM in ddH_2_O were incubated in a water bath at 100 °C for 5 h, with sampling every hour. Samples were allowed to cool down to RT prior to antioxidant activity determination.

The % retention of antioxidant activity following thermal treatment was calculated according to Jiang et al. [34] using Equation (3):% antioxidant retention = (Ai/A) × 100(3)
where Ai is the % antioxidant activity of the treated samples and A is the % antioxidant activity of the non-treated control for each sample.

The storage stability was assessed according to Ho et al. [19] with minor modifications. Samples at 75 µM in ddH_2_O were lyophilized and then the powders were exposed to air at room temperature in the dark for 14 days. Each sample was re-suspended to 75 µM in ddH_2_O and stored at −20 °C until the antioxidant activity could be determined.

The % retention of antioxidant activity following exposure to air was calculated according to Ho et al. [19] using Equation (4):% antioxidant retention = (antioxidant activity of N day/antioxidant activity of first day) × 100(4)

#### Antioxidant Determination

The antioxidant activity of the samples was determined using the ABTS assay modified according to Serem and Bester [102]. Fresh ABTS^+^ stock solution was prepared from ABTS salt by reacting 3 mM K_2_S_2_O_8_ with 8 mM ABTS in 0.1 M PBS (pH 7.4). The solution was incubated in the dark at room temperature for at least 12 h and used within 16 h after preparation. A working solution of 0.26 mM ABTS was prepared by diluting the ABTS stock solution 30× in PBS. A 30 µL volume of the sample solution was added to and mixed with 270 µL of freshly prepared ABTS working solution to wells of a 96-well plate and incubated at RT for 15 min in the dark. The absorbance was measured at 734 nm using the EMax^®^ Microplate Reader and % antioxidant activity calculated according to Jiang et al. [34] using Equation (5):% antioxidant activity = ((A − Ai)/A) × 100(5)
where A is the absorbance of the diluted ABTS^+^ with ddH_2_O blank and Ai is the absorbance when the sample was added to ABTS^+^.

### 3.6. In Vitro Digestion of the Inclusion Complexes

For this purpose, a simple digestion method as described by Daglia et al. [104] and an internationally recognized complex static in vitro simulated digestion method compiled by Minekus et al. [105] were adopted.

#### 3.6.1. Simple In Vitro Digestion

The simple digestion (SD) was carried out according to Daglia et al. [104] with minor modifications. A 20 mg/mL pepsin stock solution was prepared in 1 M HCl for gastric digestion. For the gastroduodenal digestion stock solution, 4 mg of pancreatin was dissolved in 1 mL of 1 M NaHCO_3_ solution.

To digest the sample solutions, the pH of all samples (encapsulated and non-encapsulated) was lowered to 2.5 for the gastric phase. This was followed by the addition of 5 µL pepsin per mL of sample, and the mixture was then incubated at 37 °C for 30 min with brief vortex mixing every 10 min. At the end of the gastric digestion, the pH was then increased to 6.5 for the gastroduodenal digestion phase. This was then followed by the addition of the pancreatin stock solution at 5 µL per mL of gastric digesta. The mixtures were then further incubated at 37 °C for 1 h with brief vortex mixing every 10 min. At the end of the incubation period, the digestion was terminated by the addition of Pefabloc^®^ to a final concentration of 0.1 mM per sample followed by snap freezing in liquid nitrogen. The digested samples were then centrifuged at 2504×
*g* and aliquots of the supernatant stored at −20 °C away from direct sunlight exposure until assay.

#### 3.6.2. Complex In Vitro Digestion

A three-stage complex in vitro digestion (CD) of the samples was undertaken using the method by Minekus et al. [105]. The simulated salivary fluid (SSF) comprised 15.1 mM KCl, 3.7 mM KH_2_PO_4_, 13.6 mM NaHCO_3_, 0.15 mM MgCl_2_, and 0.06 mM (NH_4_)_2_CO_3_. The pH of SSF was adjusted to 7.00 with either 1 M NaOH or 6 M HCl solutions. The simulated gastric fluid (SGF) was prepared from 6.9 mM KCl, 0.9 mM KH_2_PO_4_, 25 mM NaHCO_3_, 47.2 mM NaCl, 0.1 mM MgCl_2_, and 0.5 mM (NH_4_)_2_CO_3_. The pH of SGF was adjusted to 3.00 with either 1 M NaOH or 6 M HCl solutions. Finally, simulated intestinal fluid (SIF) was made up of 6.8 mM KCl, 0.8 mM KH_2_PO_4_, 55 mM NaHCO_3_, 38.4 mM NaCl, and 0.33 mM MgCl_2_. The pH of SIF was adjusted to 7.00 with either 1 M NaOH or 6 M HCl solution.

For oral digestion, 5 mL of each sample was mixed with 3.5 mL SSF, followed by the addition of 0.5 mL α-amylase (1500 U/mL) prepared in SSF, 25 µL of 0.3 M CaCl_2_, and 975 µL ddH_2_O. The digestion mixture was kept at 37 °C for 2 min. After the 2 min of oral digestion, 7.5 mL of pre-warmed SGF was added to the oral digest in the same vessel for simulated gastric digestion. A volume of 1.6 mL pepsin (25,000 U/mL) was added followed by 5 µL of 0.3 M CaCl_2_, 200 µL 1 M HCl, and 695 µL ddH_2_O. Where necessary, the pH of the gastric phase digestion was adjusted to 3 using 1 M HCl and the mixtures were incubated at 37 °C for 2 h with brief vortex mixing every 10 min. The intestinal digestion was commenced after the 2 h of incubation for gastric digestion by the addition of 11 mL SIF to the gastric chyme. This was followed by the addition of 5 mL of pancreatin (800 U/mL), 2.5 mL bile extract at a 160 mM concentration, 40 µL of 0.3 M CaCl_2_, 150 µL of 1 M NaOH, and 1.31 mL ddH_2_O. The pH of the resulting mixtures was adjusted to 7 with 1 M NaOH and incubated at 37 °C for 2 h with brief vortex mixing every 10 min. At the end of the intestinal digestion, Pefabloc^®^ to a final concentration of 0.1 mM was added to the digestion mixtures to stop the enzyme activity and further snap frozen in liquid nitrogen to terminate digestion. The digested samples were then centrifuged at 2504× *g* and aliquots of the supernatant stored at −20 °C away from direct sunlight exposure until assay.

### 3.7. Antioxidant Properties

#### 3.7.1. Oxygen Radical Absorbance Capacity (ORAC) Assay

The polarity of the antioxidant molecules may affect the quantified antioxidant activity depending on the assay employed [106]. As the digestive environment is aqueous and at the concentrations used the polyphenols were soluble, the ORAC assay was performed as described by Serem and Bester [102] with slight modifications. A volume of 20 µL of the sample solution was added to 155 µL of 0.139 nM fluorescein working solution in wells of a white-bottom 96-well plate. A volume of 25 µL of 0.24 M AAPH solution was added to each well containing the mixture of fluorescein and sample. The fluorescence was then measured at excitation and emission wavelengths of 485 and 520 nm, respectively, using the FLUOstar^®^ Omega Plate Reader (BMG LABTECH, Ortenberg, Germany) every 5 min for 2 h. Trolox (0.0–0.6 mM) in 0.01 M ORAC buffer was used for the standard curve and results expressed in µM TE.

#### 3.7.2. Advanced Glycation End-Products (AGEs) Assay

This method was performed according to Franco et al. [74] with minor modifications. In a sterile environment, a volume of 50 µL of the sample solution at 400 µM was added to 50 µL of 40 mg/mL BSA prepared in 0.1 M phosphate buffer, pH of 7.40, and followed by 50 µL of 56 mM MGO or FRU. Buffer was added to make a final mixture volume of 200 µL. A mixture of BSA, MGO, or FRU and buffer only comprised the positive control while the negative control was made up of BSA plus buffer only. Each mixture was prepared in a 1.5 mL sterile tube and incubated for 1 week at 37 °C. Samples were then transferred to 96-well fluorescent plates and the fluorescence was measured using the FLUOstar^®^ Omega Plate Reader at excitation and emission wavelengths of 330 and 420 nm, respectively. The results were expressed as % antiglycation and calculated from the fluorescence values using Equation (6):% antiglycation = (1 − ((sample − negative control)/(positive control − negative control))) × 100(6)

#### 3.7.3. Dichlorofluorescein Diacetate (DCFH–DA) Assay

The cellular antioxidant activity (CAA) assay was performed according to Moyo et al. [107]. The Caco-2 cells were plated at a seeding density of 5 × 10^4^ cells/mL in a 96-well plate and incubated at 37 °C and 5% CO_2_ for 24 h to allow for attachment. To 100 µL of the cells, 50 µL of 75 µM DCFH–DA was added to each well and incubated for 1 h at 37 °C. The medium was removed after incubation and cells washed once with 0.1 M PBS. Subsequently, 50 µL of the polyphenolic samples at 20 and 200 µM (final concentration of 10 and 100 µM) were added to cells in duplicate, followed by the addition of 50 µL of 8 mg/mL AAPH solution. The change in fluorescence was then measured with a FLUOstar^®^ Omega Plate Reader every 2 min for 1 h at excitation and emission wavelengths of 485 and 520 nm, respectively. Wells with the cells and buffer only served as a vehicle control while cells with AAPH only served as a positive control. The gradient of the change in the fluorescence was determined and used to calculate the CAA according to Moyo et al. [107] using Equation (7):% CAA = (1 − ((sample − vehicle control)/(positive control − vehicle control))) × 100(7)

#### 3.7.4. Cytotoxicity Assay

Following the antioxidant assay, it was necessary to assess the potential cytotoxic effects of the samples by evaluating the cell viability using the crystal violet (CV) assay as described by Delgado-Roche et al. [108] with slight modifications. This assay was performed to establish that the samples had no adverse effects on the Caco-2 cells.

### 3.8. Statistical Analysis

Results are expressed as mean ± SEM. The data was analyzed for significant differences at *p* < 0.05 using GraphPad Prism version 7.00 for Windows (GraphPad Software, La Jolla, CA, USA). Normally distributed data was analyzed with one-way ANOVA with post hoc Tukey analysis while non-normally distributed data was analyzed with the Kruskal–Wallis test with the post hoc Dunn’s multiple comparisons test. The means were ranked using RStudio (R version 4.0.3, R Foundation for statistical computing, Vienna, Austria) with Tukey multiple comparisons analysis (*p* < 0.05).

## 4. Conclusions

βCD formed 1:1 molecular inclusion complexes with CAT, GA, and EGCG and the combinations. The inclusion complexes presented a new solid phase with some improved thermal stability for encapsulated CAT and EGCG after 5 h at 100 °C and the storage stability for all encapsulated polyphenols was increased although not statistically significant for the CAT/GA/EGCG combination. Inclusion complexation did not adversely affect the antioxidant activity of the individual polyphenols and the CAT/GA/EGCG combination. Following in vitro digestion, improvements in the antiglycation and cellular antioxidant activity were observed. As part of mixtures, the activity of CAT, GA, and EGCG was enhanced.

## Figures and Tables

**Figure 1 molecules-27-03808-f001:**
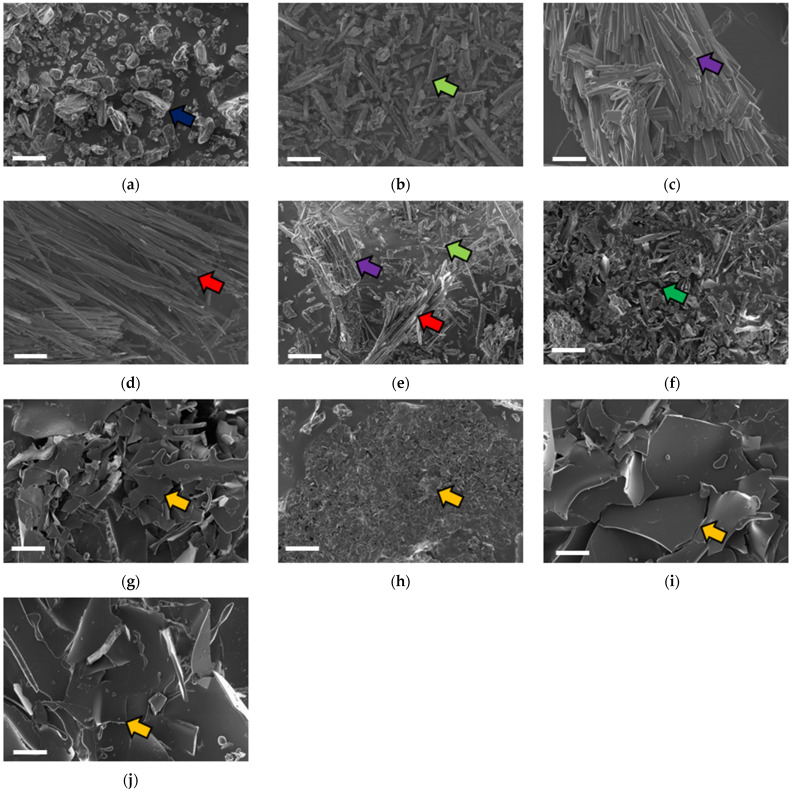
Scanning electron micrographs of pure (**a**) βCD, (**b**) CAT, (**c**) GA, (**d**) EGCG, and (**e**) CAT/GA/EGCG. Lyophilized (**f**) βCD, and lyophilized inclusion complexes of βCD with (**g**) CAT, (**h**) GA, (**i**) EGCG, and (**j**) CAT/GA/EGCG. βCD—beta cyclodextrin, CAT—catechin, GA—gallic acid, EGCG—epigallocatechin gallate. Blue, light green, purple, red, and orange arrows indicate the crystals of βCD, CAT, GA, EGCG, and inclusion complexes, respectively. The green arrow indicates the amorphous βCD. Scale bars represent 20 µm.

**Figure 2 molecules-27-03808-f002:**
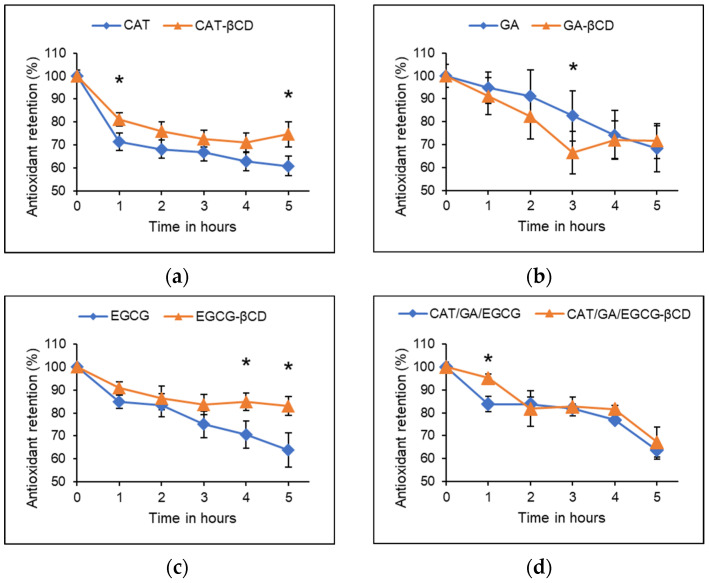
Antioxidant retention (%) following incubation at 100 °C for 5 h determined with the ABTS assay by 75 µM solutions of non- and encapsulated (**a**) CAT, (**b**) GA, (**c**) EGCG, and (**d**) CAT/GA/EGCG triple combination polyphenol samples. The data is represented as the mean ± SEM of at least three experiments done in triplicates. CAT—catechin, GA—gallic acid, EGCG—epigallocatechin gallate, βCD—beta cyclodextrin. The * represents a significant (*p* < 0.05) difference analyzed using the Kruskal–Wallis test with a post hoc Dunn’s multiple comparisons test.

**Figure 3 molecules-27-03808-f003:**
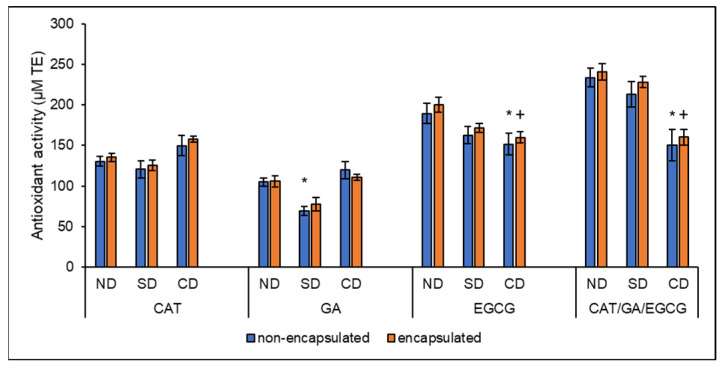
Antioxidant activity, determined with the ORAC assay of non-digested (ND), and following simple (SD) and complex (CD) digestion of non-encapsulated and encapsulated CAT, GA, and EGCG and the CAT/GA/EGCG triple combination at 100 µM. The data is represented as the mean ± SEM of at least five experiments performed in duplicates. ND—non-digested, SD—simple digestion, CD—complex digestion, ORAC—oxygen radical absorbance capacity, CAT—catechin, GA—gallic acid, EGCG—epigallocatechin gallate. The * represents significant (*p* < 0.05) differences between non-encapsulated SD or CD samples compared with the non-encapsulated ND sample. The + represents significant (*p* < 0.05) differences between encapsulated SD or CD samples compared with the encapsulated ND sample.

**Figure 4 molecules-27-03808-f004:**
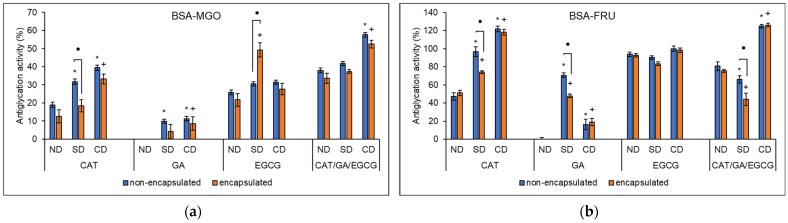
The percentage inhibition of (**a**) MGO- and (**b**) FRU-mediated AGEs formation by non-digested (ND) and following simple (SD) and complex (CD) digestion of non-encapsulated and encapsulated CAT, GA, EGCG, and the CAT/GA/EGCG combination at 100 µM for each polyphenol in each sample. The data is represented as the mean ± SEM of at least five experiments performed in duplicates. ND—non-digested, SD—simple digestion, CD—complex digestion, AGEs—advanced glycation end-products, BSA—bovine serum albumin, MGO—methylglyoxal, FRU—fructose, CAT—catechin, GA—gallic acid, EGCG—epigallocatechin gallate. The * represents significant (*p* < 0.05) differences between non-encapsulated SD or CD samples compared with the non-encapsulated ND sample. The + represents significant (*p* < 0.05) differences between encapsulated SD or CD samples compared with the encapsulated ND sample. The ● denotes significant (*p* < 0.05) difference between the non-encapsulated and encapsulated ND, SD, and CD samples.

**Figure 5 molecules-27-03808-f005:**
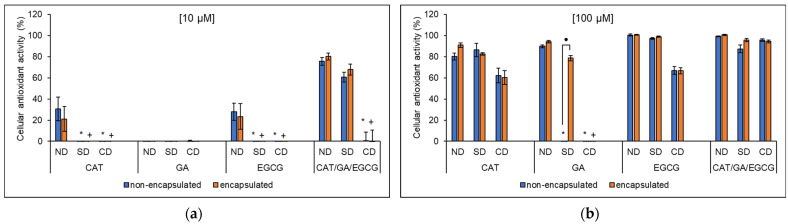
Cellular antioxidant activity of (**a**) 10 and (**b**) 100 µM non-digested (ND), and following simple (SD) and complex (CD) digestion of non-encapsulated and encapsulated CAT, GA, EGCG, and the CAT/GA/EGCG combination evaluated in the Caco-2 cell line. The data is represented as the mean ± SEM of at least five experiments performed in duplicates. ND—non-digested, SD—simple digestion, CD—complex digestion, CAT—catechin, GA—gallic acid, EGCG—epigallocatechin gallate. The * represents significant (*p* < 0.05) differences between non-encapsulated SD or CD samples compared with the non-encapsulated ND sample. The + represents significant (*p* < 0.05) differences between encapsulated SD or CD samples compared with the encapsulated ND sample. The ● denotes significant (*p* < 0.05) differences between the non-encapsulated and encapsulated ND, SD, and CD samples.

**Table 1 molecules-27-03808-t001:** Mass spectrometry parameters for parent to product transitions in both negative and positive ionization mode.

Compound (MW (g/mol))	Q_1_ Parent Ion (*m*/*z*)	Q_3_ Product Ion (*m*/*z*)	DP (V)	CE (eV)	CXP (V)
CAT (290.27)	291.4	139.1	65	25	10
		123.1	65	25	10
GA (170.12)	169.1	125.1	−30	−21	−8
		79.1	−30	−33	−13
EGCG (458.37)	459.5	289.4	54	12	10
		139.3	54	35	10
EGCG (458.37)	457.5	169.1	−51	−61	−10
		125.1	−51	−25	−10
βCD (1134.98)	1136.2	325.4	63	41	18
		487.5	63	31	10
βCD (1134.98)	1134.4	1014.4	−180	−62.5	−14
		852.3	−180	−62.5	−10
CAT + βCD (1425.25)	1424.5	1134.4	−70	−55	−10
		289.1	−70	−70	−10
GA + βCD (1305.10)	1304.5	169.1	−65	−82	−10 −10 −10
		1134.5	−65	−28	
		1219.4	−65	−52	
EGCG + βCD (1593.35)	1594.5	325.1	62	65	10
		487.0	62	55	10
		289.1	62	80	10

βCD—beta cyclodextrin, CAT—catechin, GA—gallic acid, EGCG—epigallocatechin gallate, MW—average molecular weight (g/mol), DP—declustering potential, CE—collision energy, CXP—cell collision potential.

**Table 2 molecules-27-03808-t002:** Encapsulation yield and efficiency of polyphenol inclusion complexes with β-cyclodextrin obtained via lyophilization.

Inclusion Complexes	Encapsulation Yield (%)	Encapsulation Efficiency (%)
CAT+βCD	91.27 ± 2.90 ^a^	96.62 ± 0.61 ^cd^
GA+βCD	93.36 ± 5.45 ^a^	95.65 ± 1.34 ^d^
EGCG+βCD	92.49 ± 3.25 ^a^	98.16 ± 0.56 ^cd^
CAT/GA+βCD	98.96 ± 8.89 ^a^	98.99 ± 1.63 ^bc^
CAT/EGCG+βCD	93.37 ± 5.84 ^a^	101.48 ± 1.41 ^b^
GA/EGCG+βCD	98.75 ± 7.41 ^a^	99.23 ± 0.66^b c^
CAT/GA/EGCG+βCD	94.45 ± 3.05 ^a^	104.42 ± 0.78 ^a^

The data is represented as mean ± standard error of the mean (SEM) of at least three independent repeats. βCD—beta cyclodextrin, CAT—catechin, GA—gallic acid, EGCG—epigallocatechin gallate. Different superscript letters in each column represent significant differences (*p* < 0.05).

**Table 3 molecules-27-03808-t003:** Percentage antioxidant retention, determined with the ABTS assay, of polyphenols following exposure to atmospheric oxygen in the dark at room temperature.

Sample		Antioxidant Retention (%)		
		Control	Week 1	Week 2
CAT	non-encapsulated	100 ± 1.07 ^a^	94.81 ± 2.29 ^b^	68.75 ± 1.94 ^c^
	encapsulated	100 ± 2.22 ^a^	96.06 ± 2.70 ^ab^	92.92 ± 1.92 ^b,^*
GA	non-encapsulated	100 ± 0.32 ^a^	95.63 ± 1.13 ^b^	87.61 ± 0.86 ^c^
	encapsulated	100 ± 1.18 ^a^	98.10 ± 1.33 ^ab^	96.31 ± 0.48 ^b,^*
EGCG	non-encapsulated	100 ± 0.60 ^a^	98.06 ± 1.73 ^a^	77.50 ± 0.62 ^b^
	encapsulated	100 ± 0.90 ^a^	92.99 ± 3.91 ^b^	85.44 ± 3.83 ^c,^*
CAT/GA/EGCG	non-encapsulated	100 ± 0.56 ^b^	104.77 ± 0.68 ^a^	93.22 ± 0.97 ^c^
	encapsulated	100 ± 1.33 ^a^	95.79 ± 1.26 ^b,^*	96.64 ± 1.53 ^b^

The data is represented as mean ± SEM of at least three experiments performed in triplicates. CAT—catechin, GA—gallic acid, EGCG—epigallocatechin gallate. Different superscript letters in each row represent significant differences (*p* < 0.05). The * denotes a significant difference (*p* < 0.05) in antioxidant retention between non-encapsulated and encapsulated samples of each compound for each week.

## Data Availability

Some data is available as Appendix A and the rest under the UPSpace institutional repository, URI: http://hdl.handle.net/2263/84453 (date dissertation was available online 11 March 2022).

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
