# Peer review of "Stability, Morphology, and Effects of In Vitro Digestion on the Antioxidant Properties of Polyphenol Inclusion Complexes with β-Cyclodextrin"

_molecules, 2022, doi:10.3390/molecules27123808_

Round 1
Reviewer 1 Report
The manuscript studied the encapsulation of CAT, GA, EGCG and the CAT/GA/EGCG triple combination with β-CD, and the morphology and stability and the effects of in vitro digestion on the antioxidant properties of the encapsulation complexes. Overall, the manuscript was well organized and wroten and its data are sufficient. However, there are some suggestions that the authors should be concerned.
1. The novelty in the present study is very limited because there are many studies have been conducted to explore the main content of this manuscript. The authors have compared the results with those reported in previous studies, however, there are no subversive conclusions were found.
2. Retention of antioxidant activity following thermal treatment and exposure to atmospheric oxygen for two weeks were analyzed to reveal the thermal stability and storage stability of the smaples, respectively. It's not convincing enough because the degradation products had higher antioxidant activity relative to native forms and the increase in activity could be attributed to synergistic effects between neoformed and native molecules.
3. Correlation analysis between CAT, GA and EGCG and the CAT/GA/EGCG triple combination in terms of the stability and antioxidant properties of the encapsulation complexes should be added.
Author Response
|
The manuscript studied the encapsulation of CAT, GA, EGCG and the CAT/GA/EGCG triple combination with β-CD, and the morphology and stability and the effects of in vitro digestion on the antioxidant properties of the encapsulation complexes. Overall, the manuscript was well organized and wroten and its data are sufficient. However, there are some suggestions that the authors should be concerned. |
|
Comment 1: The novelty in the present study is very limited because there are many studies have been conducted to explore the main content of this manuscript. The authors have compared the results with those reported in previous studies, however, there are no subversive conclusions were found. |
|
Response: Shortly the novelty of this study is related the 1) the polyphenol combination used, 2) the method used for the encapsulation of these polyphenols and 3) the effects of digestion. To support the novelty of the study, an extensive literature search was undertaken and only the following related studies were identified. Related to the polyphenols used in the present study and β-CD encapsulation: Xu et al. (2019), reported that with β-CD encapsulation the stability of EGCG was increased and with encapsulation inhibition of osteoclastogenesis by EGCG was increased. Roy et al. (2017) reported that β-CD encapsulation of gallic acid, EGC and EGCG did not prevent the formation dityrosine cross linkages associated with oxidative stress but did prevent protein oligomerization. No study has been undertaken to further investigate the effects of β-CD encapsulation on the antioxidant properties of gallic acid, catechin and EGCG. Related to the effects of digestion: In a review article: Role of the Encapsulation in Bioavailability of Phenolic Compounds by Grgíc, et al., 2020, based on 285 literature references, the authors identified the following regarding the in vitro and ex vivo effects of polyphenol encapsulation. For gallic acid: No studies have investigated the effects of encapsulation on the activity of gallic acid For catechin hydrate (CAT): Two studies investigated the release, ex vivo permeation and pharmacodynamics of CAT (phosphatidylcholine liposomes), and the in vivo and bioactivity retention and antidiabetic properties of CAT (horse chestnut, water chestnut and lotus stem starch with freeze drying). For EGCG: Two studies that investigated the retention/release of EGCG (gum arabic- maltodextrin) and stability of EGCG (chitosan-triphosphate) in the mouse GIT. For mixtures that contain the above polyphenols such as green tea or green tea polyphenols different encapsulation methods, but not β-CD have been used and the effects on in vitro digestion has been investigated on epithelial permeability and intestinal transport. The effects of in vitro digestion on antioxidant properties is lacking. No recent 2022 publication was identified that addressed this gap in knowledge. This information has been added to the manuscript. |
|
Comment 2: Retention of antioxidant activity following thermal treatment and exposure to atmospheric oxygen for two weeks were analyzed to reveal the thermal stability and storage stability of the smaples, respectively. It's not convincing enough because the degradation products had higher antioxidant activity relative to native forms and the increase in activity could be attributed to synergistic effects between neoformed and native molecules. |
|
Response: We agree with this comment and the above has been clarified. Lines 330 – 334. |
|
Comment 3: Correlation analysis between CAT, GA and EGCG and the CAT/GA/EGCG triple combination in terms of the stability and antioxidant properties of the encapsulation complexes should be added. |
|
Response: As conditions used for the stability studies differ from those used for the determination of antioxidant activity it is not possible to correlate the stability and antioxidant properties. |
Reviewer 2 Report
Author should Improve title. Word “The” is repeating and overal its little bit confusing.
In abstract what do you mean by simple and complex method of digestion.
Which encapsulation material was used.
In Introduction Line 33 and 34 Sentence “The use of natural products such as nutraceuticals rich in polyphenols for potential NCDs prevention and treatment limits the need for pharmaceutical interventions that are often associated with adverse effects” is unclear. Nutraceuticals are not a product, and what will be adverse effects.
In line 48 author can mention and explain some encapsulation techniques.
In Line 57 and 58 The sentence is not correct and it is almost against your hypothesis please rewrite it. Di-gestion and rapid metabolism of polyphenols limits the bioaccessibility which inherently affects bioavailability and activity.
Results and Discussion has been given at heading 2 it should come after materials and methods. Please correct it. Always Methodology come before the results and discussion.
Results are confusing explain it in more better way along with their discussion.
First paragraph of Materials and Methods you can write raw material were purchased from the given location.
In Line 623 heading 3.3 Author should must mention Condition and details of Encapsulator and encapsulation material. Authors can also give details of encapsulation method.
In heading 3.5.1 author have checked Thermal and Storage stability, while in results you have only discuss antioxidant. You can add some more parameters to assess the storage stability.
Statistical analysis is quiet good but you can refine it.
There are too many grammatical mistakes that need serious attentions.
Recheck all the references according to the journal format.
Author Response
|
Comment 1: Author should Improve title. Word “The” is repeating and overal its little bit confusing. |
|
Response: The title has been changed in the manuscript to read as follows, “Stability, morphology and effects of in vitro digestion on the antioxidant properties of polyphenol inclusion complexes with β-cyclodextrin” |
|
Comment 2: In abstract what do you mean by simple and complex method of digestion. |
|
Response: This information has been added to the abstract. |
|
Comment 3: Which encapsulation material was used. |
|
The current study used beta-cyclodextrin encapsulation material. The polyphenols were encapsulated in solution into beta-cyclodextrin through molecular inclusion complexation and the inclusion complexes were obtained in powder form through lyophilisation. This was clarified throughout the manuscript. |
|
Comment 4: In Introduction Line 33 and 34 Sentence “The use of natural products such as nutraceuticals rich in polyphenols for potential NCDs prevention and treatment limits the need for pharmaceutical interventions that are often associated with adverse effects” is unclear. Nutraceuticals are not a product, and what will be adverse effects. |
|
Response: This has been corrected and more detail has been provided. Line 34-38. |
|
Comment 5: In line 48 author can mention and explain some encapsulation techniques. |
|
Response: The sentence has been revised to read as follows “A wide range of encapsulation techniques such as lyophilisation, spray-drying, coacervation and emulsification with polymer wall materials such as βCD, maltodextrin, pectin and sodium alginate suitable for different active compounds are available.” |
|
Comment 6: In Line 57 and 58 The sentence is not correct and it is almost against your hypothesis please rewrite it. Digestion and rapid metabolism of polyphenols limits the bioaccessibility which inherently affects bioavailability and activity. |
|
Response: Revised as advised. |
|
Comment 7: Results and Discussion has been given at heading 2 it should come after materials and methods. Please correct it. Always Methodology come before the results and discussion. |
|
Response: The template used was according to the instructions to authors of the journal. |
|
Comment 8: Results are confusing explain it in more better way along with their discussion. |
|
The entire manuscript has been edited to better explain the results and the associated discussion. |
|
Comment 9: First paragraph of Materials and Methods you can write raw material were purchased from the given location. |
|
Response: This section has been edited, and shorted, and now provides information on the source of key reagents, enzymes, media and the cell line used. |
|
Comment 10: In Line 623 heading 3.3 Author should must mention Condition and details of Encapsulator and encapsulation material. Authors can also give details of encapsulation method. |
|
Response: The methods have been revised to include the details and condition of the encapsulator under heading 3.1. The details of encapsulation method and encapsulation material are discussed under heading “3.1. Preparation of inclusion complexes” |
|
Comment 11: In heading 3.5.1 author have checked Thermal and Storage stability, while in results you have only discuss antioxidant. You can add some more parameters to assess the storage stability. |
|
Response: We agree with the reviewer, but as the focus of this study was the antioxidant properties, the stability related to antioxidant activity was evaluated. |
|
Comment 12: Statistical analysis is quiet good but you can refine it. |
|
Response: The section on statistical analysis has been edited and refined. |
|
Comment 13: There are too many grammatical mistakes that need serious attentions. |
|
Response: All grammatical errors have been corrected. |
|
Comment 14: Recheck all the references according to the journal format. |
|
Response: Citations and references were according to MDPI.ens references style file from the EndNote website. |
Reviewer 3 Report
This study investigating encapsulation molecules using beta-cyclodextrin for polyphenols was carefully performed and of interest for many readers.
However, an important consideration is lacking. As described by the authors, some polyplenols have difficulty with water solubility. Because living body consists of lipophilic and hydrophilic materials, evaluation of antioxidant activity depending on lipophilic and hydorophilic conditions is quite important (Methods Enzymol. 1992; 213:460-72. doi: 10.1016/0076-6879(92)13148-q.). This viewpoint should be discussed in antioxidant determination section.
Author Response
|
This study investigating encapsulation molecules using beta-cyclodextrin for polyphenols was carefully performed and of interest for many readers. |
|
Comment 1: However, an important consideration is lacking. As described by the authors, some polyplenols have difficulty with water solubility. Because living body consists of lipophilic and hydrophilic materials, evaluation of antioxidant activity depending on lipophilic and hydorophilic conditions is quite important (Methods Enzymol. 1992; 213:460-72. doi: 10.1016/0076-6879(92)13148-q.). This viewpoint should be discussed in antioxidant determination section. |
|
Response: This consideration was included in the methods – see ORAC assay. |
Round 2
Reviewer 1 Report
The authors explain my concerns, and the manuscript can be accepcted in present form.